# Unraveling the Roots of Income Polarization in Europe: A Divided Continent

Michele Fabiani [ID]

Department of Political Science, Communication and International Relations, University of Macerata, Via Don Minzoni 22/A, 62100 Macerata, Italy; m.fabiani9@unimc.it

**Abstract:** The issue of polarization, as opposed to inequality, has been little explored in European countries. In this paper, using data provided by the Luxembourg Income Studies Database, we look at the trend of income polarization in 12 European countries, the only ones available with two comparable years, using the relative distribution method. The results clearly show a trend toward polarization in almost the cases analyzed, with a concentrated prevalence in the lower tail of the distribution, thus observing a worsening in the distribution. Next, we look at drivers that may have contributed to these changes, using the RIF-regression method. It is interesting to observe how these characteristics are in many cases common across all countries: the occupational sector, level of education and area of residence have the same impact, albeit with different intensities, in all countries. This suggests the possibility of coordinated intervention across these nations, acting on the same variables for all of them.

**Keywords:** Europe; income polarization; RIF-regression; relative distribution

**JEL Classification:** C14; D31; D63

## 1. Introduction

Income polarization as a notion is comparatively understudied, particularly in Europe, compared to inequality and poverty, which have both gotten a lot of attention in the literature. Although both categories are sensitive to the middle of the distribution, income polarization and inequality are two distinct phenomena. While income inequality focuses on how far apart various members of a society are from the general mean, income polarization contrasts homogeneity within a group with the overall variability of a particular community (Castro 2003). Income polarization therefore resembles segregation more than income disparity (Esteban and Ray 1994).

The middle class is dwindling, which may be related to income polarization. Every civilization needs a prosperous middle class since it is linked to high income, rapid economic expansion, and social and political stability (Easterly 2001; Pressman 2007). High income polarization, on the other hand, suggests a divided society and may result in the creation of social conflict, discontent and tension (Esteban and Ray 1994, 1999; Gradín 2000; Zhang and Kanbur 2001). Although both income inequality and income polarization indicate shifts in the middle of the income distribution, income polarization is more likely to result in social unrest and political disorder.

In addition to causing societal discontent and conflict, income polarization can also have other negative effects. First of all, less social mobility results from a highly income-polarized society due to the fact that it may be challenging for the comparatively poor to advance up the income scale (Motiram and Sarma 2014). Polarization of income also has a negative impact on economic growth (Brzeziński 2013; Ezcurra 2009), affecting redistribution with possible negative effects, for example, on consumption. One explanation is that the social unrest and political unpredictability that underlie income polarization

could adversely affect market operations, labor relations, and the security of property rights (Keefer and Knack 2002). Moreover, income polarization is detrimental to health because it lowers the availability of some public goods and increases psychosocial stress due to social friction and conflict (Pérez and Ramos 2010).

This article contributes to the literature by observing the polarization trends in 12 European countries for the period from the early 2000s to the end of the second decade of the century. It provides a different view than previous works for two main reasons: it observes income distribution through polarization analysis instead of using traditional inequality measures, and it analyzes how household characteristics impact the distribution. To observe these trends, this paper uses the "relative distribution" method (Handcock and Morris 1998, 1999), a non-parametric approach. This methodology has not been used very frequently for the analysis of polarization and has been limited to single-country cases in Europe. This paper aims to provide a more complete picture of polarization analysis in the region, showing updated and harmonized data across countries. Furthermore, the relative distribution method is able to provide us with results that are easily usable and of immediate interpretation.

As a consequence, within the relative distribution framework, the paper applies a novel methodology to identify the covariates of distributional changes. The methodology used, i.e., the RIF-regression proposed by Firpo et al. (2009), is able to observe the impact of covariates on the distribution in detail, providing the possibility to assess economic policy interventions that should be taken to counteract the phenomenon of income polarization.

The paper is organized as follows: Section 2 provides a literature review on the issue of polarization in Europe; Section 3 discusses the data and provides summary statistics; Section 4 outlines the distinctive features of the relative distribution approach and presents the proposed RIF-regression approach; Section 5 details the main findings of the study; Section 6 provides summary conclusions.

## 2. Literature Review

Income polarization, defined as a divergence of income levels in a population, has been a topic of great interest in recent years, particularly outside Europe. Scholars have examined the matter in countries such as China (Araar 2008; Zhang and Kanbur 2001), India (Chakravarty and Majumder 2001; Motiram and Sarma 2014), Nigeria (Awoyemi and Araar 2009; Clementi et al. 2015), Sub-Saharan Africa (Clementi et al. 2019, 2021, 2022a) and Latin American countries (Deutsch et al. 2014; Gasparini et al. 2008), as well as in more developed countries like the United States and Canada (D'Ambrosio and Wolff 2001; Foster and Wolfson 1992, 2010). While there has been some research on income polarization in Europe, studies focused on this topic are relatively rare, and little attention has been paid to income polarization in new member states of Central and Eastern Europe (CEE NMS).

Most studies on income polarization in Europe use different approaches than the one proposed in this paper, in addition to analyzing the conditions of individual countries. Some of them use the polarization index DER, which identifies the formation of poles on the basis of within- and between-group effects. It is observed that in Italy and Spain (D'Ambrosio 2001; Gradín 2000), the specific characteristics of workers, such as education or area of residence, significantly influence the formation of these poles, thus leading to an increase in polarization. A similar approach has been used to analyze income polarization in Denmark. According to this study (Hussain 2009) the main factor that has led to an increase in this phenomenon is the between-group component, with a growing alienation of the people, who distance themselves more and more from each other, also bringing more inequality. For Poland (Brzeziński 2011), using the DER bi-polarization index, significant differences can be observed between the trend of inequality and that of polarization; the main component of the increase in polarization is found, in fact, in the growing identification within the groups themselves, leading to different trends between inequality and polarization.

Instead of analyzing the Italian situation, Poggi and Silber (2010) used the polarization index proposed by Deutsch et al. (2007); they showed that when taking the identity of

the individuals into account, a distinction can be made between a change over time in polarization and a change in polarization. The first one is the consequence of the so called "structural mobility", i.e., the change over time in the overall, between- and within-group inequality. The second one is the sole consequence of "exchange mobility", i.e., the changes over time in the ranks of individuals.

Analyzing the case of Germany, Gigliarano and Mosler (2009) used a multidimensional approach in the analysis of polarization, which no longer focuses on income polarization only, but also on education. The polarization index in this case will be a function of three components: inequality within groups, inequality between groups and the size of the groups themselves.

Atkinson and Brandolini (2013) used various measures of inequality to observe how changes in the distribution impact the middle class: a greater polarization on the tails of the distribution leads to a progressive emptying of the middle class, with both economic and social negative consequences. The results show that this effect is greater for southern European countries than for continental and northern European ones.

Finally, the relative distribution method has been used in some cases to observe income polarization in Russia and Italy (Nissanov and Pittau 2016; Massari et al. 2009b; Ricci and Scicchitano 2021). Studies show how during the analyzed periods, polarization grows in the country, even significantly. However, unlike our approach, these papers do not investigate what might be the driver that impacts on the changes in distribution leading to the observed growing polarization of income.

## 3. Data and Summary Statistics

In this paper, data are taken from the Luxembourg Income Study Database (LIS)[1]. LIS acquires datasets with income, wealth, employment and demographic data from many high- and middle-income countries, and harmonizes them to enable cross-national comparisons. Data used cover 12 European countries[2] available in the dataset and for which it is possible to have comparable surveys for two separate years.

The variable used in the first part for the distribution analysis is household-disposable income, net of income taxes and contributions[3].

To analyze the impact of social conditions on polarization trends in the second part of the paper, demographic, geographic, employment status and educational level of the head of household variables are used.

The period of analysis covers about two decades for all countries. Before turning to the analysis of polarization using the relative distribution method, it is interesting to look at some statistics regarding the trend of income inequality in the countries analyzed. Starting with income trends, is has been observed that the variable has shown growth, both for different income classes and on average among all the countries considered, except Spain and Italy. In fact, these countries show a stagnation of incomes, which remain practically stationary during the period taken into analysis.

It is also very interesting to observe the trend in the Gini index and the Foster-Wolfson polarization index, except in some specific cases, such as Germany and Luxembourg, which show a significant increase in inequality. For the remaining countries, the change is minimal or absent. In the United Kingdom, we even observe a decrease in these indexes.

## 4. Methodology
*4.1. Polarization and Relative Distribution*

In the analysis of income distribution, the topic of polarization has gained significance over the past 20 years (Foster and Wolfson 1992; Esteban and Ray 1994; Wolfson 1994, 1997), and it now appears that polarization is widely recognized as a separate concept from inequality.

Regardless of where a community is positioned along the income scale, a broad definition of income polarization (Esteban and Ray 1994) describes it as the "clustering" of a population around two or more poles of distribution. In a multi-group setting, the idea

of income polarization aims to quantify the degree of potential conflict present in a given distribution (see Esteban and Ray 1999, 2008, 2011). The concept is to think of society as a collection of groups, where members of one group have traits in common with one another (i.e., a sense of "identification"), but vary from members of other groups (i.e., a sense of "alienation") in terms of the same traits.

Therefore, political or social conflict is more likely the more homogeneous and separate the groups are; that is, when the within-group income distribution is more concentrated around its local mean, the between-group income distance is greater (see, inter alia, Gradín 2000; Milanovic 2000; D'Ambrosio 2001; Zhang and Kanbur 2001; Montalvo and Reynal-Querol 2002; Duclos et al. 2004; Lasso de la Vega and Urrutia 2006; Esteban et al. 2007; Gigliarano and Mosler 2009; Poggi and Silber 2010).

While summary measures of income polarization are frequently used in literature, a different (yet non-parametric) approach has emerged to measure the growth of the middle class and the level of household income polarization in a number of middle- and high-income nations. This approach is known as the "relative distribution" and combines the strengths of summary polarization indexes with the details of distributional change provided by the kernel density estimates.

The relative distribution method has been employed by Alderson et al. (2005), Massari et al. (2009a, 2009b), Alderson and Doran (2011), Borraz et al. (2013), Clementi and Schettino (2013, 2015), Clementi et al. (2015, 2017, 2019, 2021, 2022a), Molini and Paci (2015), Petrarca and Ricciuti (2016), and Nissanov and Pittau (2016).

More formally,[4] let $Y_0$ be the income variable for the reference population and $Y$ the income variable for the comparison population. The relative distribution is defined as the ratio of the density of the comparison population to the density of the reference population, evaluated at the relative data $r$:

$$g(r) = \frac{f\left(F_0^{-1}(r)\right)}{f_0\left(F_0^{-1}(r)\right)} = \frac{f(y_r)}{f_0(y_r)}, \ 0 \leq r \leq 1, \ y_r \geq 0, \tag{1}$$

where $f(\cdot)$ and $f_0(\cdot)$ denote the density functions of $Y$ and $Y_0$, respectively, and $y_r = F_0^{-1}(r)$ is the quantile function of $Y_0$. When no changes occur between the two distributions, $g(r)$ has a uniform distribution. A value of $g(r)$ higher (lower) than 1 means that the share of households in the comparison population is higher (lower) than the corresponding share in the reference population at the $r$th quantile of the latter.

One of the major advantages of this method is the possibility to decompose the relative distribution into changes in location and shape. The decomposition can be written as:

$$\underbrace{\frac{f(y_r)}{f(y_r)}}_{Overall} = \underbrace{\frac{f_{0L}(y_r)}{f_0(y_r)}}_{Location} \times \underbrace{\frac{f(y_r)}{f_{0L}(y_r)}}_{Shape}. \tag{2}$$

$F_{0L}(y_r)$ is the median-adjusted density function:

$$f_{0L}(y_r) = f_0(y_r + \rho), \tag{3}$$

where the value $\rho$ is the difference between the medians of the comparison and reference distributions; alternative indexes like the mean and/or multiplicative location shift can also be considered.

The relative distribution approach also includes a median relative polarization index, which is a measurement of the degree to which the comparison distribution is more polarized than the reference one:

$$MRP = \frac{4}{n} \left( \sum_{i=1}^{n} \left| r_i - \frac{1}{2} \right| \right) - 1. \tag{4}$$

The MRP index can be additively decomposed into contributions to the overall polarization made by the lower and upper halves of the median-adjusted relative distribution, enabling one to distinguish downgrading from upgrading. In terms of data, the lower relative polarization (LRP) index and the upper relative polarization (URP) index can be calculated as shown in Equations (5) and (6):

$$LRP = \frac{8}{n} \left[ \sum_{i=1}^{n/2} \left( \frac{1}{2} - r_i \right) \right] - 1. \tag{5}$$

$$URP = \frac{8}{n} \left[ \sum_{i=\frac{n}{2}+1}^{n} \left( r_i - \frac{1}{2} \right) \right] - 1. \tag{6}$$

with $MRP = \frac{1}{2}(LRP + URP)$. The *MRP*, *LRP* and *URP* indexes range from $-1$ to 1, and equal 0 when there is no change.

### 4.2. RIF-Regression Model

To analyze the drivers of income polarization in countries under consideration, the Recentered Influence Function (RIF) regression was used. The strength of the correlation between modest change in one covariate and change in a relative polarization index (such as MRP, LRP or URP) can be calculated using this method.

The influence function (Cowell and Victoria-Feser 1996) captures the effects of explanatory variables on the distributional statistic of interest, while also reflecting the influence of a single observation on a given distributional statistic, such as a particular quantile.

Firpo et al. (2009) proposed a simple modification in which the quantile is added back to the influence function, resulting in what the authors call the "re-centered influence function" (RIF):

$$RIF(y_i; q_\tau, F_Y) = q_\tau + IF(y_i; q_\tau, F_Y),$$

where $q_\tau$ is the $\tau$-th quantile of the distribution of household incomes $Y$, and $IF(\cdot)$ is the influence function. With this result, Firpo et al. (2009) show that we can model the conditional expectation of the RIF as a simple linear function of explanatory variables.

In practice, following Firpo et al.'s (2009) procedure, one can first obtain an estimate of the RIF for each income *i* by using Equation (6) above; then, the following equation can be estimated using an ordinary least-squares method (OLS):

$$RIF(y_i; \hat{q}_\tau, F_Y) = \alpha_\tau + \sum_{k=1}^{K} \beta_{\tau,k} \cdot x_{\tau,i,k} + \varepsilon_{\tau,i}, \quad i = 1, 2, \dots, N,$$

where $\alpha_\tau$ is a constant, $x_{\tau,i,k}$ denotes a realization of the *k*-th explanatory variable, $\beta_{\tau,k}$ is the corresponding coefficient and $\varepsilon_{\tau,i}$ is the corresponding error term. The estimated model parameters $\hat{\beta}_{\tau,k}$, termed "unconditional quantile partial effect", can be interpreted as the effect of a small change in the distribution of $X_k$ on the quintile $q_\tau$—when the distribution of other covariates remains unchanged—or as a linear approximation of the effect of large changes of $X_k$ on $q_\tau$ (e.g., Firpo et al. 2018).

Although Firpo et al. (2009) initially concentrated on the analysis of partial eects of explanatory variables on unconditional quantiles of the dependent variable, the underlying ideas of this methodology have been applied to other distributional statistics (for example, see Essama-Nssah and Lambert 2012; Rios-Avila 2020; Jann 2021)[5].

## 5. Results
### 5.1. Relative Distribution Results

There are two important aspects to consider. As noted in Table 1, the results obtained with this method are contrasted with the inequality results obtained with traditional measures (e.g., Gini index), where we do not observe a clear and common trend across countries. This underscores the importance of looking at the income distribution from dif-

ferent vantage points as well, to capture changes in the distribution, as relative distribution does. Relative distribution indexes show a homogenous pattern throughout the European countries surveyed. MRP, LRP and URP indexes are specified in Table 2, and graphs for each country, that allow an immediate and easy-to-read view of changes in distribution, are available in Appendix A. All are undergoing an accentuated polarization process, as evidenced by the positive and significant value of the MRP. The second aspect to consider is in which part of the distribution this concentration occurs. In 9 out of 11 countries (we exclude Italy from the analysis, where the index values are non-significant), the LRP index value is higher than the URP value, showing a more pronounced concentration in the lower tail than in the upper tail. This aspect is very significant, because it leads to a general worsening of the income distribution on the one hand, and on the other hand, to a progressive emptying of the middle class, which, as the results show, is sucked into the lower tail of the distribution.

**Table 1.** Summary statistics and inequality and polarization indexes.

| Country | Year [a] | P10 | P25 | P50 | Mean | P75 | P90 | Gini | Fgt0 | FW |
|---|---|---|---|---|---|---|---|---|---|---|
| Austria | 2000 | 15,453.1 | 21,222.0 | 28,213.5 | 30,736.8 | 36,628.5 | 48,616.9 | 25.4 | 13.7 | 20.1 |
| | 2019 | 17,059.0 | 25,105.6 | 34,097.5 | 37,783.3 | 46,174.6 | 61,489.0 | 27.4 | 15.4 | 21.7 |
| Belgium | 2000 | 14,104.1 | 18,805.4 | 27,002.7 | 30,313.4 | 35,868.5 | 47,013.6 | 28.8 | 16.2 | 22.0 |
| | 2017 | 15,335.6 | 21,104.6 | 30,880.9 | 32,847.5 | 40,884.2 | 51,391.1 | 26.0 | 18.4 | 22.1 |
| Denmark | 2000 | 15,953.2 | 20,758.6 | 28,254.5 | 29,686.8 | 35,772.6 | 43,914.9 | 22.5 | 13.1 | 18.2 |
| | 2016 | 17,606.8 | 22,771.4 | 31,255.8 | 34,168.6 | 41,277.4 | 52,309.4 | 25.5 | 12.8 | 20.5 |
| Finland | 2000 | 12,605.8 | 16,324.7 | 22,272.2 | 24,495.9 | 28,995.4 | 36,976.2 | 25.3 | 12.7 | 20.0 |
| | 2016 | 15,827.0 | 20,916.9 | 28,139.0 | 31,381.4 | 37,410.7 | 48,046.0 | 25.8 | 12.6 | 20.4 |
| France | 2000 | 12,744.5 | 17,415.9 | 24,065.7 | 27,858.2 | 33,224.6 | 46,376.4 | 29.4 | 14.9 | 23.6 |
| | 2018 | 13,815.4 | 19,275.9 | 26,970.8 | 31,095.5 | 36,605.6 | 50,458.9 | 30.2 | 16.0 | 23.0 |
| Germany | 2000 | 15,184.2 | 20,825.9 | 27,538.6 | 30,606.1 | 36,915.0 | 48,383.3 | 25.9 | 12.5 | 20.7 |
| | 2019 | 15,291.0 | 22,182.4 | 31,483.6 | 35,222.0 | 42,200.6 | 56,566.9 | 29.3 | 17.2 | 22.8 |
| Ireland | 2000 | 9322.0 | 14,279.8 | 22,236.7 | 25,007.8 | 31,176.6 | 41,731.6 | 31.3 | 22.5 | 26.4 |
| | 2019 | 16,008.1 | 21,494.2 | 30,308.7 | 34,853.2 | 42,441.0 | 55,680.9 | 28.7 | 15.5 | 23.8 |
| Italy | 2000 | 8945.6 | 13,425.4 | 20,400.7 | 23,793.6 | 29,665.8 | 40,105.7 | 33.4 | 20.1 | 28.2 |
| | 2016 | 8206.8 | 12,741.7 | 19,503.7 | 22,359.3 | 28,518.8 | 39,064.8 | 33.9 | 21.1 | 29.1 |
| Luxembourg | 2000 | 21,288.6 | 27,628.0 | 37,282.0 | 42,403.1 | 51,239.9 | 69,413.8 | 26.2 | 12.3 | 22.8 |
| | 2019 | 21,967.5 | 30,451.9 | 43,198.6 | 49,813.8 | 61,648.5 | 82,427.8 | 29.6 | 16.4 | 25.5 |
| Netherland | 1999 | 15,596.0 | 20,104.8 | 26,656.0 | 28,670.3 | 34,764.7 | 43,537.3 | 23.1 | 11.1 | 19.0 |
| | 2018 | 16,802.5 | 22,378.2 | 30,833.0 | 34,284.5 | 41,330.4 | 53,688.4 | 27.0 | 13.8 | 21.6 |
| Spain | 2000 | 9857.0 | 14,486.5 | 22,246.8 | 26,265.5 | 32,556.0 | 46,205.8 | 33.7 | 20.8 | 29.2 |
| | 2016 | 8852.7 | 14,606.6 | 23,047.6 | 26,407.8 | 34,173.7 | 46,516.1 | 34.1 | 22.6 | 29.6 |
| United Kingdom | 2000 | 10,543.7 | 14,622.7 | 22,152.3 | 27,690.4 | 32,969.3 | 47,166.8 | 35.7 | 20.3 | 29.5 |
| | 2020 | 14,228.5 | 19,148.3 | 27,222.5 | 31,741.7 | 38,876.6 | 54,271.9 | 30.5 | 15.5 | 25.8 |

Source: Authors' calculations based on LIS data. [a] The time frame used for different countries is determined by data availability.

**Table 2.** Polarization indexes by country.

| Country | Index [a] | Value | LB [b] | UB [c] | *p*-Value [d] |
|---|---|---|---|---|---|
| Austria | MRP | 0.150 | 0.113 | 0.187 | 0.000 |
| | LRP | 0.161 | 0.099 | 0.222 | 0.000 |
| | URP | 0.139 | 0.091 | 0.188 | 0.000 |
| Belgium | MRP | 0.102 | 0.065 | 0.139 | 0.000 |
| | LRP | 0.148 | 0.084 | 0.212 | 0.000 |
| | URP | 0.056 | 0.003 | 0.109 | 0.036 |

**Table 2.** *Cont.*

| Country | Index [a] | Value | LB [b] | UB [c] | *p*-Value [d] |
|---|---|---|---|---|---|
| Denmark | MRP | 0.127 | 0.120 | 0.134 | 0.000 |
| | LRP | 0.101 | 0.089 | 0.113 | 0.000 |
| | URP | 0.153 | 0.143 | 0.163 | 0.000 |
| Finland | MRP | 0.155 | 0.134 | 0.177 | 0.000 |
| | LRP | 0.157 | 0.117 | 0.197 | 0.000 |
| | URP | 0.153 | 0.126 | 0.181 | 0.000 |
| France | MRP | 0.065 | 0.055 | 0.075 | 0.000 |
| | LRP | 0.105 | 0.089 | 0.121 | 0.000 |
| | URP | 0.026 | 0.014 | 0.038 | 0.000 |
| Germany | MRP | 0.148 | 0.127 | 0.168 | 0.000 |
| | LRP | 0.213 | 0.178 | 0.247 | 0.000 |
| | URP | 0.082 | 0.054 | 0.111 | 0.000 |
| Ireland | MRP | 0.120 | 0.070 | 0.170 | 0.000 |
| | LRP | 0.090 | 0.001 | 0.180 | 0.047 |
| | URP | 0.150 | 0.088 | 0.211 | 0.000 |
| Italy | MRP | −0.005 | −0.038 | 0.028 | 0.772 |
| | LRP | 0.007 | −0.054 | 0.068 | 0.823 |
| | URP | −0.017 | −0.058 | 0.024 | 0.416 |
| Luxembourg | MRP | 0.178 | 0.130 | 0.227 | 0.000 |
| | LRP | 0.234 | 0.146 | 0.321 | 0.000 |
| | URP | 0.122 | 0.065 | 0.179 | 0.000 |
| Netherland | MRP | 0.167 | 0.141 | 0.193 | 0.000 |
| | LRP | 0.188 | 0.143 | 0.232 | 0.000 |
| | URP | 0.146 | 0.111 | 0.181 | 0.000 |
| Spain | MRP | 0.045 | 0.016 | 0.075 | 0.002 |
| | LRP | 0.081 | 0.030 | 0.132 | 0.002 |
| | URP | 0.010 | −0.025 | 0.046 | 0.569 |
| United Kingdom | MRP | 0.058 | 0.030 | 0.085 | 0.000 |
| | LRP | 0.083 | 0.035 | 0.132 | 0.001 |
| | URP | 0.032 | 0.002 | 0.061 | 0.031 |

Source: Authors' calculations based on LIS data. Notes: [a] MRP = median relative polarization index; LRP = lower relative polarization index; URP = upper relative polarization index. [b] Lower bound of the 95% confidence interval. [c] Upper bound of the 95% confidence interval. [d] Refers to the null hypothesis of no change with respect to the reference distribution, i.e., that the index equals to 0.

*5.2. RIF-Regression Results*

In this section are presented the results of RIF-regressions for the three polarization indexes and different independent variables. The independent variables are divided into different categories, such as the sector of employment, education level, country of birth and area of residence. The choice of variables is related to their availability within the surveys and following what has been carried out in previous analyses of income polarization (e.g., Ezcurra 2009). Each category has a base group, against which the other groups are compared. The coefficients and standard errors for each independent variable are presented in the tables. Asterisks next to a coefficient indicate the level of statistical significance of that variable: *** denotes significance at the 1% level, ** at the 5% level and * at the 10% level.

Table 3 presents the results of a RIF-regression analysis, with the MRP index as the dependent variable and several independent variables. The independent variables are divided into different categories, and each category has a base group against which the other groups are compared. The results indicate that the sector of employment, education level, country of birth, area of residence and age have significant effects on the median relative polarization in some countries. For example, being employed in the agricultural sector has a positive effect on the median relative polarization in Austria and Belgium,

while the industry sector has a negative effect in some countries. Education level has also a significant effect, with those having a high level of education showing a positive effect on the median relative polarization in all countries. The country of birth and area of residence also have significant effects, with being born outside the country and living in rural areas having negative effects on the median relative polarization.

Table 4 shows the results of a regression analysis with the dependent variable being the lower relative polarization index of income, and several independent variables are used to explain the variation in the dependent variable across different sectors and countries. The independent variables are grouped into four categories: sector, education, country of birth and area. The coefficients of these binary variables indicate the effect of being in that subcategory on the dependent variable compared to the base category.

Table 5 displays the results of a regression analysis where the dependent variable is the upper relative polarization index of income, and various independent variables are examined across different countries. The results show that workers in the agricultural sector have a positive and significant effect on the upper relative polarization index of income compared to the not-employed group, while workers in the industry and service sectors have a negative and significant effect. Education level also has a significant effect, with workers having a high education level showing a positive and significant effect on the index compared to those with a low education level. Being born outside the country has a negative and significant effect on the index, while living in rural areas has a negative and significant effect on the index.

**Table 3.** Rif-regression results, MRP index.

| | Austria | Belgium | Denmark | Finland | France | Germany | Ireland | Italy | Luxembourg | Netherland | Spain | U.K. |
|---|---|---|---|---|---|---|---|---|---|---|---|---|
| Sector | | | | | | | | | | | | |
| Not employed | (base) | (base) | (base) | (base) | (base) | (base) | (base) | (base) | (base) | (base) | (base) | (base) |
| Agriculture | 0.090 ** | 0.095 ** | 0.926 *** | 0.171 *** | 0.029 | −0.217 | 0.047 * | 0.131 *** | 0.021 | 0.195 *** | 0.078 * | 0.088 |
| | (0.040) | (0.039) | (0.247) | (0.050) | (0.173) | (0.214) | (0.025) | (0.043) | (0.053) | (0.061) | (0.041) | (0.313) |
| Industry | 0.004 | −0.002 | 0.656 *** | 0.066 ** | −0.327 *** | −0.062 | 0.005 | 0.000 | 0.008 | −0.096 *** | −0.034 * | 0.360 *** |
| | (0.018) | (0.014) | (0.084) | (0.029) | (0.072) | (0.043) | (0.013) | (0.033) | (0.016) | (0.028) | (0.020) | (0.059) |
| Services | 0.002 | −0.015 | 0.672 *** | 0.033 | −0.306 *** | −0.021 | 0.008 | −0.010 | 0.025 ** | −0.049 *** | −0.021 | 0.218 *** |
| | (0.012) | (0.010) | (0.050) | (0.022) | (0.057) | (0.029) | (0.009) | (0.024) | (0.012) | (0.016) | (0.016) | (0.042) |
| Education | | | | | | | | | | | | |
| Low | (base) | (base) | (base) | (base) | (base) | (base) | (base) | (base) | (base) | (base) | (base) | (base) |
| Medium | −0.001 | −0.019 * | −0.242 *** | −0.057 ** | 0.230 *** | −0.150 *** | 0.039 *** | 0.098 *** | 0.015 | −0.020 | 0.099 *** | 0.064 |
| | (0.013) | (0.009) | (0.050) | (0.024) | (0.052) | (0.035) | (0.010) | (0.019) | (0.010) | (0.016) | (0.016) | (0.047) |
| High | 0.080 *** | 0.053 *** | 1.127 *** | 0.200 *** | 2.229 *** | 0.209 *** | 0.106 *** | 0.358 *** | 0.096 *** | 0.176 *** | 0.309 *** | 0.469 *** |
| | (0.017) | (0.009) | (0.057) | (0.025) | (0.062) | (0.039) | (0.009) | (0.034) | (0.011) | (0.017) | (0.015) | (0.043) |
| Country of birth | | | | | | | | | | | | |
| Born in the country | (base) | (base) | | | (base) | (base) | (base) | (base) | (base) | (base) | (base) | (base) |
| Born outside the country | −0.014 | 0.017 *** | | | 0.335 *** | 0.039 | −0.055 *** | −0.173 *** | −0.016 * | 0.074 *** | −0.020 | −0.041 |
| | (0.013) | (0.010) | | | (0.061) | (0.033) | (0.009) | (0.040) | (0.009) | (0.023) | (0.020) | (0.0549) |
| Area | | | | | | | | | | | | |
| Cities | (base) | (base) | (base) | (base) | (base) | | (base) | | (base) | | (base) | |
| Towns and suburbs | −0.008 | −0.021 *** | −0.650 *** | −0.108 *** | −0.461 *** | | −0.024 ** | | −0.022 | | −0.057 *** | |
| | (0.012) | (0.008) | (0.052) | (0.021) | (0.076) | | (0.010) | | (0.013) | | (0.015) | |
| Rural areas | −0.031 ** | −0.034 *** | −0.609 *** | −0.117 *** | −0.570 *** | | −0.041 *** | | −0.017 | | −0.102 *** | |
| | (0.012) | (0.011) | (0.081) | (0.022) | (0.087) | | (0.008) | | (0.013) | | (0.013) | |
| Age | 0.001 *** | −0.000 | 0.009 *** | 0.001 | 0.010 *** | −0.003 *** | 0.001 *** | 0.000 | 0.001 ** | 0.000 | 0.001 * | 0.000 |
| | (0.000) | (0.000) | (0.001) | (0.001) | (0.001) | (0.001) | (0.000) | (0.000) | (0.000) | (0.000) | (0.000) | (0.001) |

**Table 3.** *Cont.*

| | Austria | Belgium | Denmark | Finland | France | Germany | Ireland | Italy | Luxembourg | Netherland | Spain | U.K. |
|---|---|---|---|---|---|---|---|---|---|---|---|---|
| Sex | | | | | | | | | | | | |
| Male | (base) | (base) | (base) | (base) | (base) | (base) | (base) | (base) | (base) | (base) | (base) | (base) |
| Female | 0.016 (0.010) | 0.002 (0.007) | 0.219 *** (0.048) | −0.074 *** (0.018) | −0.329 *** (0.049) | −0.066 *** (0.025) | −0.003 (0.007) | −0.053 *** (0.016) | −0.003 (0.009) | −0.040 *** (0.014) | −0.034 *** (0.012) | 0.155 *** (0.036) |
| N° Household members | −0.001 (0.004) | −0.000 (0.003) | −0.125 *** (0.020) | −0.053 *** (0.007) | −0.016 (0.019) | −0.050 *** (0.009) | −0.003 (0.002) | 0.019 ** (0.008) | −0.007 ** (0.003) | −0.014 *** (0.005) | 0.014 *** (0.004) | −0.067 *** (0.016) |

Notes: Robust standard errors in brackets; * $p < 0.10$, ** $p < 0.05$, *** $p < 0.01$.

**Table 4.** Rif-regression results, LRP index.

| | Austria | Belgium | Denmark | Finland | France | Germany | Ireland | Italy | Luxembourg | Netherland | Spain | U.K. |
|---|---|---|---|---|---|---|---|---|---|---|---|---|
| Sector | | | | | | | | | | | | |
| Not employed | (base) | (base) | (base) | (base) | (base) | (base) | (base) | (base) | (base) | (base) | (base) | (base) |
| Agriculture | 0.118 ** (0.059) | 0.123 (0.077) | 1.906 *** (0.427) | 0.267 *** (0.088) | −0.633 ** (0.298) | −0.477 (0.408) | 0.093 ** (0.042) | 0.167 ** (0.076) | −0.007 (0.095) | 0.236 ** (0.103) | 0.081 (0.076) | −0.193 (0.489) |
| Industry | 0.004 (0.030) | 0.022 (0.024) | 2.232 *** (0.151) | 0.152 *** (0.051) | −0.634 *** (0.122) | −0.080 (0.075) | 0.026 (0.024) | 0.061 (0.061) | 0.020 (0.032) | −0.095 * (0.051) | 0.019 (0.037) | 0.744 *** (0.105) |
| Services | −0.010 (0.020) | 0.012 (0.018) | 2.139 *** (0.090) | 0.046 (0.040) | −0.853 *** (0.096) | 0.008 (0.047) | 0.025 (0.016) | 0.055 (0.046) | 0.056 ** (0.023) | −0.068 ** (0.028) | −0.011 (0.029) | 0.311 *** (0.079) |
| Education | | | | | | | | | | | | |
| Low | (base) | (base) | (base) | (base) | (base) | (base) | (base) | (base) | (base) | (base) | (base) | (base) |
| Medium | −0.004 (0.024) | −0.001 (0.017) | 0.025 (0.096) | −0.067 (0.045) | 0.479 *** (0.097) | −0.196 *** (0.064) | 0.066 *** (0.019) | 0.152 *** (0.038) | 0.032 (0.020) | −0.055 * (0.029) | 0.166 *** (0.031) | 0.103 (0.088) |
| High | 0.063 ** (0.030) | 0.088 *** (0.017) | 2.069 *** (0.104) | 0.245 *** (0.045) | 2.401 *** (0.104) | 0.225 *** (0.068) | 0.145 *** (0.018) | 0.374 *** (0.054) | 0.158 *** (0.020) | 0.180 *** (0.029) | 0.390 *** (0.027) | 0.617 *** (0.079) |
| Country of birth | | | | | | | | | | | | |
| Born in the country | (base) | (base) | | | (base) | (base) | (base) | (base) | (base) | (base) | (base) | (base) |
| Born outside the country | −0.019 (0.023) | −0.025 (0.019) | | | 0.354 *** (0.105) | 0.018 (0.059) | −0.092 *** (0.017) | −0.344 *** (0.075) | −0.056 *** (0.016) | 0.066 (0.040) | −0.097 ** (0.039) | −0.132 (0.098) |

**Table 4.** *Cont.*

| | Austria | Belgium | Denmark | Finland | France | Germany | Ireland | Italy | Luxembourg | Netherland | Spain | U.K. |
|---|---|---|---|---|---|---|---|---|---|---|---|---|
| Area | | | | | | | | | | | | |
| Cities | (base) | (base) | (base) | (base) | (base) | | (base) | | (base) | | (base) | |
| Towns and suburbs | −0.010 (0.019) | −0.022 (0.015) | −0.464 *** (0.089) | −0.083 ** (0.038) | −0.638 *** (0.141) | | −0.035 ** (0.017) | | −0.028 (0.023) | | −0.059 ** (0.028) | |
| Rural areas | −0.057 *** (0.019) | −0.037 * (0.020) | −0.192 *** (0.146) | −0.079 ** (0.040) | −0.780 *** (0.156) | | −0.047 *** (0.015) | | −0.013 (0.024) | | −0.123 *** (0.026) | |
| Age | 0.000 (0.000) | −0.001 ** (0.000) | 0.000 (0.002) | −0.001 (0.001) | −0.004 * (0.002) | −0.005 *** (0.001) | 0.001 ** (0.000) | 0.000 (0.001) | 0.001 *** (0.001) | −0.003 *** (0.000) | 0.001 (0.001) | −0.001 (0.002) |
| Sex | | | | | | | | | | | | |
| Male | (base) | (base) | (base) | (base) | (base) | (base) | (base) | (base) | (base) | (base) | (base) | (base) |
| Female | 0.013 (0.016) | −0.014 (0.013) | 0.079 (0.088) | −0.077 ** (0.032) | −0.496 *** (0.086) | −0.047 (0.045) | 0.001 (0.013) | −0.040 (0.031) | 0.004 (0.017) | −0.089 *** (0.025) | −0.044 * (0.022) | −0.226 *** (0.066) |
| N° Household members | −0.002 (0.006) | 0.006 (0.005) | 0.114 *** (0.034) | −0.079 *** (0.012) | −0.045 (0.034) | −0.045 *** (0.016) | −0.005 (0.004) | 0.006 (0.014) | −0.011 (0.007) | −0.013 (0.009) | 0.029 *** (0.008) | −0.108 *** (0.029) |

Notes: Robust standard errors in brackets; * $p < 0.10$, ** $p < 0.05$, *** $p < 0.01$.

**Table 5.** Rif-regression results, URP index.

| | Austria | Belgium | Denmark | Finland | France | Germany | Ireland | Italy | Luxembourg | Netherland | Spain | U.K. |
|---|---|---|---|---|---|---|---|---|---|---|---|---|
| Sector | | | | | | | | | | | | |
| Not employed | (base) | (base) | (base) | (base) | (base) | (base) | (base) | (base) | (base) | (base) | (base) | (base) |
| Agriculture | 0.062 (0.054) | 0.067 * (0.040) | −0.054 (0.353) | 0.075 (0.068) | 0.692 *** (0.191) | 0.041 (0.163) | 0.000 (0.033) | 0.094 ** (0.041) | 0.049 (0.046) | 0.154 * (0.084) | 0.075 * (0.043) | 0.371 (0.232) |
| Industry | 0.005 (0.026) | −0.027 (0.019) | −0.920 *** (0.119) | −0.020 (0.035) | −0.019 (0.089) | −0.044 (0.058) | −0.015 (0.018) | −0.062 (0.040) | −0.002 (0.014) | −0.097 ** (0.043) | −0.088 *** (0.027) | −0.023 (0.072) |
| Services | 0.014 (0.015) | −0.043 *** (0.012) | −0.794 *** (0.067) | 0.019 (0.025) | 0.240 *** (0.064) | −0.051 (0.044) | −0.008 (0.011) | −0.077 *** (0.024) | −0.004 (0.012) | −0.030 (0.021) | −0.031 (0.019) | 0.125 *** (0.042) |

**Table 5.** *Cont.*

| | Austria | Belgium | Denmark | Finland | France | Germany | Ireland | Italy | Luxembourg | Netherland | Spain | U.K. |
|---|---|---|---|---|---|---|---|---|---|---|---|---|
| Education | | | | | | | | | | | | |
| Low | (base) | (base) | (base) | (base) | (base) | (base) | (base) | (base) | (base) | (base) | (base) | (base) |
| Medium | 0.002 (0.015) | −0.037 *** (0.012) | −0.510 *** (0.067) | −0.048 * (0.027) | −0.019 (0.058) | −0.104 ** (0.052) | 0.012 (0.012) | 0.044 ** (0.022) | −0.001 (0.010) | 0.015 (0.021) | 0.032 (0.020) | 0.025 (0.052) |
| High | 0.097 *** (0.021) | −0.017 (0.012) | 0.186 ** (0.078) | 0.155 *** (0.029) | 2.057 *** (0.073) | 0.193 *** (0.058) | 0.066 *** (0.011) | 0.342 *** (0.036) | 0.034 *** (0.012) | 0.171 *** (0.023) | 0.229 *** (0.019) | 0.322 *** (0.048) |
| Country of birth | | | | | | | | | | | | |
| Born in the country | (base) | (base) | | | (base) | (base) | (base) | (base) | (base) | (base) | (base) | (base) |
| Born outside the country | −0.010 (0.016) | 0.061 *** (0.012) | | | 0.317 *** (0.067) | 0.060 (0.045) | −0.017 * (0.010) | −0.001 (0.031) | 0.022 * (0.011) | 0.082 *** (0.026) | 0.056 ** (0.023) | 0.050 (0.056) |
| Area | | | | | | | | | | | | |
| Cities | (base) | (base) | (base) | (base) | (base) | | (base) | | (base) | | (base) | |
| Towns and suburbs | −0.007 (0.017) | −0.020 * (0.011) | −0.836 *** (0.072) | −0.134 *** (0.025) | −0.284 *** (0.085) | | −0.014 (0.012) | | −0.016 (0.015) | | −0.055 *** (0.019) | |
| Rural areas | −0.005 (0.016) | −0.031 ** (0.015) | −1.025 *** (0.115) | −0.156 *** (0.026) | −0.360 *** (0.107) | | −0.035 *** (0.010) | | −0.021 (0.015) | | −0.082 *** (0.018) | |
| Age | 0.001 *** (0.001) | 0.001 * (0.001) | 0.019 *** (0.001) | 0.002 *** (0.001) | 0.025 *** (0.001) | −0.001 (0.001) | 0.001 * (0.000) | 0.001 ** (0.001) | 0.000 (0.000) | 0.002 *** (0.001) | 0.001 ** (0.000) | 0.001 (0.001) |
| Sex | | | | | | | | | | | | |
| Male | (base) | (base) | (base) | (base) | (base) | (base) | (base) | (base) | (base) | (base) | (base) | (base) |
| Female | 0.018 (0.012) | 0.019 * (0.011) | 0.359 *** (0.068) | −0.071 *** (0.021) | −0.162 *** (0.050) | −0.086 ** (0.035) | −0.006 (0.009) | −0.066 *** (0.020) | −0.010 (0.011) | 0.008 (0.018) | −0.025 (0.016) | −0.084 ** (0.039) |
| N° Household members | −0.000 (0.005) | −0.008 * (0.004) | −0.364 *** (0.028) | −0.026 *** (0.009) | 0.012 (0.022) | −0.055 *** (0.014) | −0.001 (0.003) | 0.033 *** (0.008) | −0.003 (0.003) | −0.016 ** (0.006) | −0.001 (0.006) | −0.027 (0.016) |

Notes: Robust standard errors in brackets; * $p < 0.10$, ** $p < 0.05$, *** $p < 0.01$.

## 6. Discussion

The research presented is intended to provide new insights from previous analyses of inequality and polarization in Europe. Starting precisely with the distinction between the analysis of income inequality and polarization, the paper adds to the scant literature present for the latter issue (D'Ambrosio 2001; Gradín 2000; Hussain 2009; Brzeziński 2011; Poggi and Silber 2010). Further, it observes polarization using a method (i.e., relative distribution) hitherto sparsely employed for European countries, with few exceptions (Petrarca and Ricciuti 2016; Nissanov and Pittau 2016), which allows us to observe how incomes are distanced from each other from an absolute rather than relative point of view (Clementi et al. 2022b), as shown by other indexes proposed in previous research.

Turning to the second part of the analysis, the comparability of the country surveys used and the innovative RIF-regression method (Firpo et al. 2009) allow us to propose a general European picture of the causes of the polarization seen earlier. Such a comparison has not been performed in such detail before, opening up possible discussions in this regard. For one, the method used allows us to observe the impact of the variables considered on income distribution in a more timely manner. Second, having carried out the same analysis for different European countries, the results open up a policy discussion, leading to an analysis of what might be common policies to counteract this growing income distancing observed previously.

## 7. Conclusions

Income polarization is a concept that is gaining increasing importance in the analysis of income distribution in Europe. Although often confused with income inequality, income polarization is a distinct phenomenon that focuses on the homogeneity within a group rather than the differences between groups. Reduction in the middle class is a significant factor that may contribute to income polarization. High levels of income polarization can lead to social unrest, political instability and economic downturns. This article has used the relative distribution method and econometric decomposition to examine the trends in income polarization in 12 European countries over the past two decades. The analysis has identified the main drivers of polarization and their impact on observable and unobservable characteristics. Data used in the analysis were obtained from the Luxembourg Income Study Database (LIS), and cover income, wealth, employment and demographic details. The paper has contributed to the literature by providing a granular analysis of distributional changes that an analysis based on standard inequality decompositions would not allow. The results of the analysis show that polarization has increased over the past two decades in most countries examined, and the main drivers of this trend have been changes in the labor market and the level of education. The implications of these findings are significant, as income polarization has far-reaching effects on society, economy and individual well-being. A highly polarized society can result in the creation of social conflict, discontent and tension. It can also lead to lack of social mobility, lower economic growth, and can have a negative impact on health outcomes. Policies aimed at reducing income polarization are therefore necessary to promote social and political stability, economic growth and individual well-being. The results of this paper confirm what has been seen in previous studies regarding polarization in specific European countries: the phenomenon is primarily driven by the different conditions of households in the labor market (Gigliarano and Mosler 2009; Brzeziński 2013). This can thus be seen as the area where most action needs to be taken to ensure better income distribution. The most important difference between this paper and previous ones concerns the methodology used; relative distribution allows us to observe differences in absolute terms (Clementi et al. 2022a), thus ensuring a different point of view and one that may open up new discussions. It is very important to find common aspects between countries which, although on the same continent, are very different from a social, economic and dimensional point of view. The results clearly show the need to intervene first and above all, in the labor market, guaranteeing greater participation and support to people in difficulty or in a state of unemployment. The second

fundamental point of intervention is the education system: a higher level of education guarantees the possibility of accessing a higher level of income. On one hand, this implies guaranteeing new generations the possibility of improving and increasing their knowledge. On the other hand, it allows those already in the labor market and with lower levels of education the possibility to increase their skills through specific trainings focused on the emerging needs and trends of the labor market. Further research is needed to explore the complex relationships between income polarization, inequality and poverty, and to effectively identify policy measures to address these issues.

**Funding:** This research received no external funding.

**Informed Consent Statement:** Informed consent was obtained from all subjects involved in the study.

**Data Availability Statement:** The Luxembourg Income Study Database (https://www.lisdatacenter.org/, accessed on 10 March 2023) provides remote access to the microdata through a web-based Job Submission Interface (LISSY). Users have to register to the platform and submit through the LISSY interface their statistical programs written in R, SAS, SPSS or Stata.

**Conflicts of Interest:** The authors declare no conflict of interest.

## Appendix A

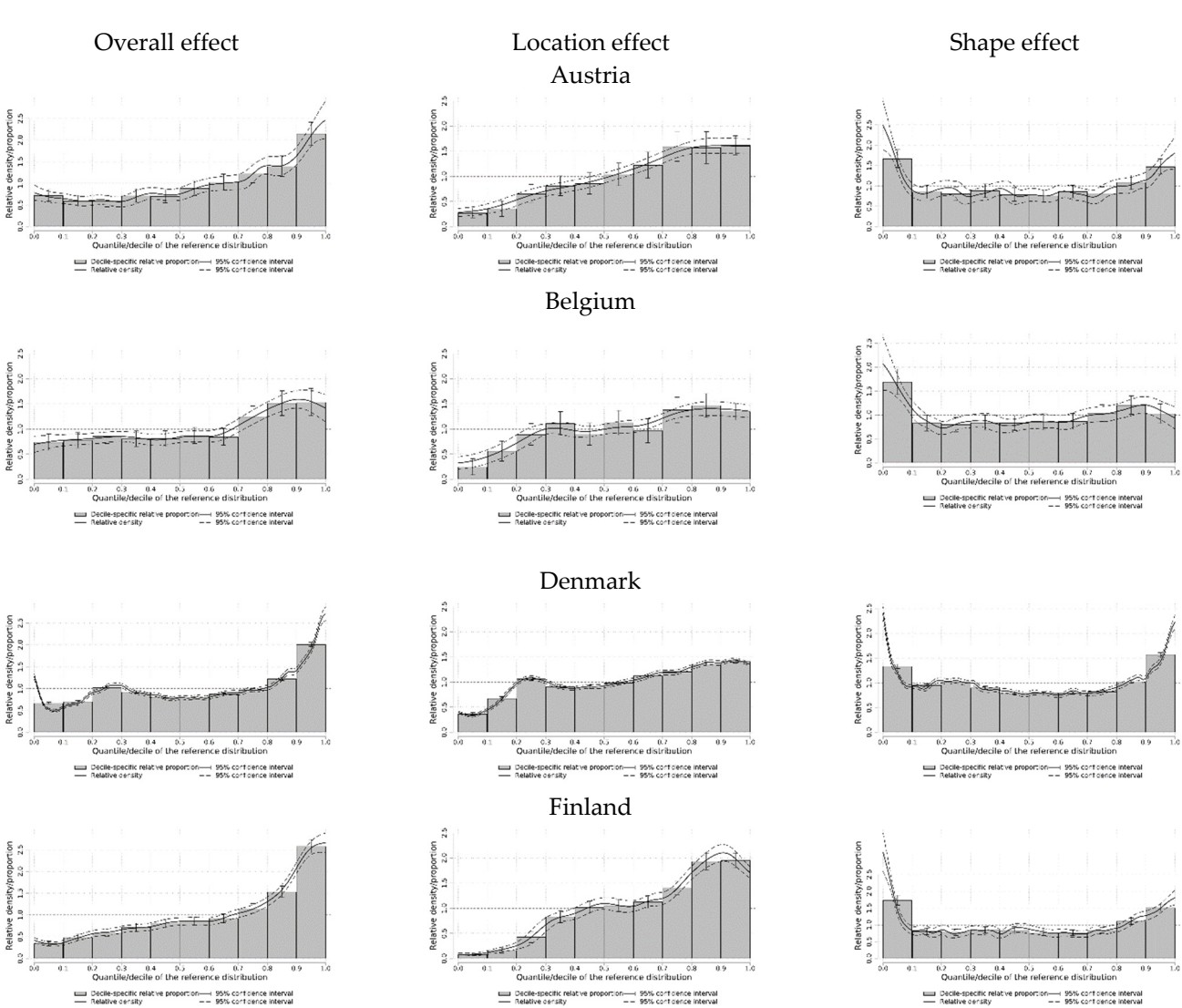

**Figure A1.** *Cont.*

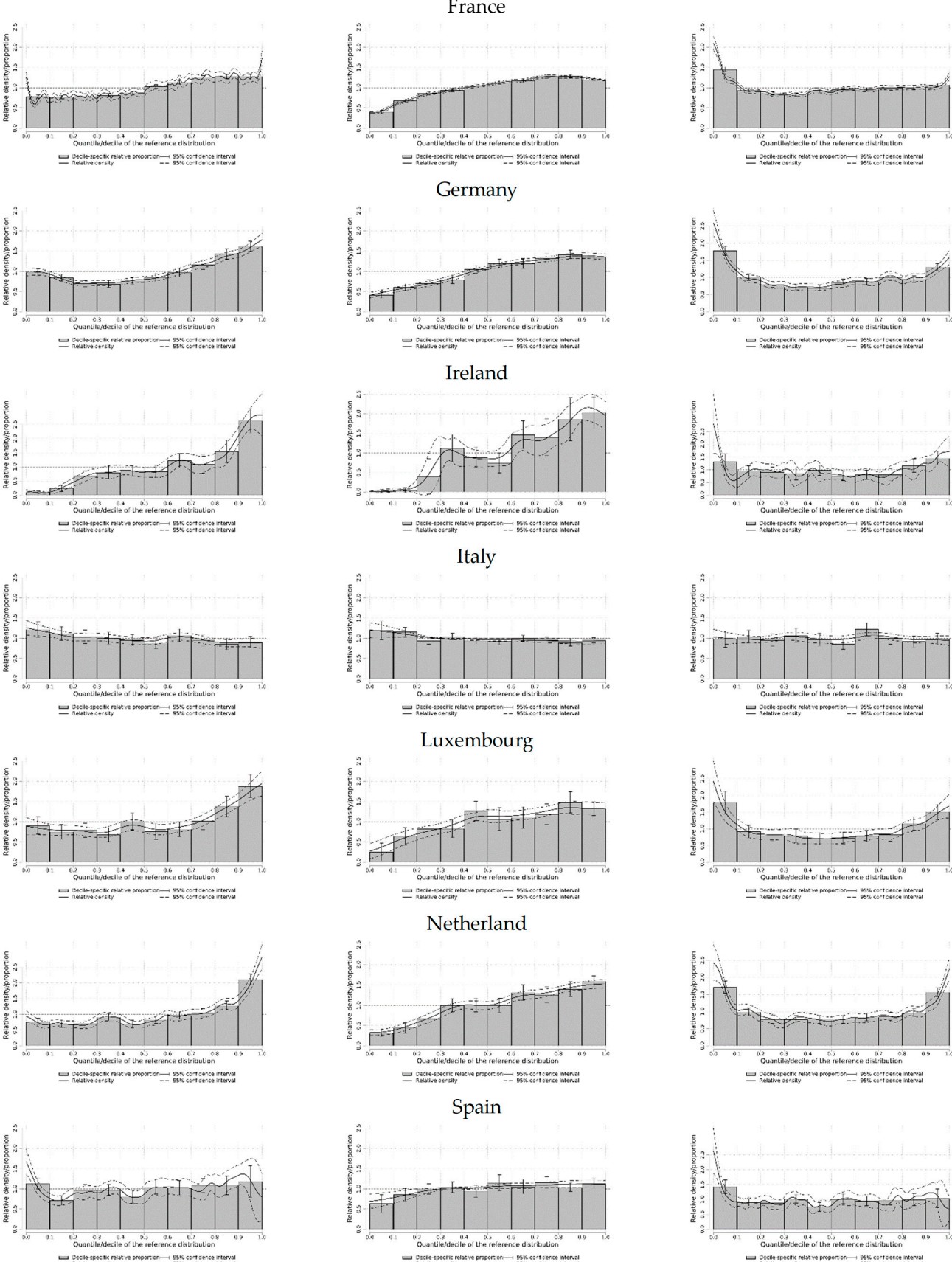

**Figure A1.** *Cont.*

United Kingdom

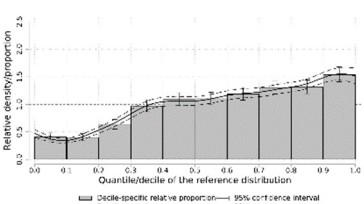
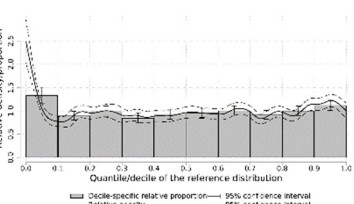

**Figure A1.** Source: Authors' calculations based on LIS data.

## Notes

[1] Luxembourg Income Study (LIS) Database, http://www.lisdatacenter.org (Accessed on 10 March 2023) (multiple countries; December 2022–January 2023). Luxembourg: LIS.

[2] Austria, Belgium, Denmark, Finland, France, Germany, Ireland, Italy, Luxembourg, Netherland, Spain and United Kingdom.

[3] "Disposable household income" is usually the preferred measure for income distribution analysis, as it is the income available to households to support their consumption expenditure and savings during the reference period (Canberra Group 2011). According to the LIS documentation (https://www.lisdatacenter.org/data-access/key-figures/methods/disposable/) Accessed on 10 March 2023, this measure includes income received from work, wealth and direct government benefits, such as retirement or unemployment benefits. The measure then subtracts direct taxes paid, such as income taxes.

[4] Here, we limit ourselves to illustrating the basic concepts behind the use of the relative distribution method. Interested readers are referred to Handcock and Morris (1998, 1999) for a more detailed explication.

[5] For a more specific observation of the use of RIF-regression applied to polarization indexes, see (Jann 2021).

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
