# Peer review of "Unraveling the Roots of Income Polarization in Europe: A Divided Continent"

_economies, doi:10.3390/economies11080217_

Round 1

Reviewer 1 Report

I would like to suggest the following recommendations for the improvement of the current manuscript:

1. The Abstract must be reworked and expanded. The author should address the relevance of the issue, existing gaps in studies/policies, how the study aims to address them, which methods are used, major findings, and potential implications of the findings. As brief and focused as possible, but still above three sentences (currently)

2. Introduction: the aim of the study must be articulated

3. Earlier studies should be discussed in a critical manner. Currently, the section is just a summary of descriptions of papers, not connected, not summarized. A problem-oriented narrative is needed here, with a summary of gaps in studies and an explanation of how the author's study attempts to bridge those gaps. 

4. Results must be discussed, the author's findings must be compared with those obtained by other scholars. The Discussion section is now missing, which radically reduces the contribution of the study to the literature

The quality of the English language and style must be improved 

Reviewer 2 Report

Congratulations on the topic of the article. This analysis complements studies on income inequality and adds useful information for policy-making.

The empirical analysis is robust, but some authors' choices must be better explained.

I recommend publication after slight changes.

Attached is the commented article.

Author Response

Regarding the point about the Italian situation, a country-specific paper is about to be released, where the same methodology was used for the country analysis.
For the remaining issues, please see the attachment

Round 2

Reviewer 1 Report

Despite certain revisions made by the author, I still consider the discussion competent to be extremely weak. I do not think mixing a conclusion with a discussion benefits this study. They must be separated, and the demonstration of the novelty of the study and the author's contributions to the literature must be improved substantially

Needs thorough proofreading to improve the quality and style
